# The Diagnostic Value of Circulating Biomarkers and Role of Drug-Coated Balloons for In-Stent Restenosis in Patients with Peripheral Arterial Disease

**DOI:** 10.3390/diagnostics12092207

**Published:** 2022-09-12

**Authors:** Nunzio Montelione, Vincenzo Catanese, Antonio Nenna, Mohamad Jawabra, Emanuele Verghi, Francesco Loreni, Francesco Nappi, Mario Lusini, Ciro Mastroianni, Federica Jiritano, Giuseppe Filiberto Serraino, Pasquale Mastroroberto, Francesco Alberto Codispoti, Massimo Chello, Francesco Spinelli, Francesco Stilo

**Affiliations:** 1Vascular Surgery, Università Campus Bio-Medico di Roma, 00128 Rome, Italy; 2Cardiac Surgery, Università Campus Bio-Medico di Roma, 00128 Rome, Italy; 3Cardiac Surgery, Centre Cardiologique du Nord de Saint-Denis, 93200 Paris, France; 4Cardiovascular Surgery, Magna Graecia University of Catanzaro, 88100 Catanzaro, Italy

**Keywords:** in-stent restenosis, drug-eluting balloon, drug-coated balloon, peripheral arterial disease

## Abstract

Peripheral arterial disease (PAD) is an increasingly pathological condition that commonly affects the femoropopliteal arteries. The current fashionable treatment is percutaneous transluminal angioplasty (PTA), often with stenting. However, the in-stent restenosis (ISR) rate after the stenting of the femoropopliteal (FP) district remains high. Many techniques have been proposed for the treatment of femoropopliteal ISR, such as intravascular brachytherapy, laser atherectomy, second stenting and drug-coated balloons angioplasty (DCB). DCB showed a significantly lower rate of restenosis and target lesions revascularization (TLR) compared to conventional PTA. However, further studies and multi-center RCTs with dedicated long-term follow-up are needed to verify the true efficiency of this approach. Nowadays, the correlation between PAD and inflammation biomarkers is well known. Multiple studies have shown that proinflammatory markers (such as C-reactive proteins) and the high plasma levels of microRNA could predict the outcomes after stent placement. In particular, circulating microRNA-320a, microRNA-3937, microRNA-642a-3p and microRNA-572 appear to hold promise in diagnosing ISR in patients with PAD, but also as predictors of stent patency. This narrative review intends to summarize the current knowledge on the value of circulating biomarkers as predictors of ISR and to foster the scientific debate on the advantages of using DCB in the treatment of ISR in the FP district.

## 1. Introduction

Peripheral arterial disease (PAD) is becoming an increasingly common pathology, with an increase in the world average age and an estimated incidence of 200 million cases per year [1,2]. PAD remains one of the most frequent manifestations of atherosclerosis and cardiovascular pathologies, together with coronary artery and cerebrovascular disease. PAD includes the stenosis of non-coronary and non-cerebral arteries and shares classic risk factors for cardiovascular disease [3]. Classical disease can involve the aorto-iliac, femoro-popliteal district and the below-the-knee arteries. Generally, the below-the-knee arteries are more frequently involved in diabetic patients. The most commonly affected district is the femoropopliteal (FP), which can cause claudication or ischemia of the lower limbs, ischemic ulcers or non-revascularizable conditions, leading to critical limb ischemia (CLI) and a risk of amputation. In the past, open surgery was generally indicated for native arterial disease, mainly bypassed with autologous grafts or with prosthetic material, but with the advent of new technologies, the endovascular approach has become predominant. Nowadays, percutaneous transluminal angioplasty (PTA) represents the first line approach of treatment for obstructive disease of the superficial femoral and popliteal artery. Often, the procedural treatment of this district requires stent implantation after PTA for residual flap dissection or very calcified residual stenosis. However, in-stent restenosis (ISR) after stenting of the FP district remains a daunting problem, resulting in a less than 50% patency rate after 3 years [4]. ISR is believed to be due to neointimal hyperplasia caused by post-PTA endothelial damage [5,6], but similarly to coronary stent disease, other factors are involved such as adherence to antiplatelet therapy [7,8] and specific stent factors [9]. The incidence of ISR also varies according to the type of stent used: 19–35% at 1 year with a grooved tube nitinol stent, and 14–17% with a twisted-wire nitinol stent [4]. The anatomical position can also determine ISR (flexion, stretching, etc.). In the past, there was also stent fracturing, which is particularly common with older self-expanding stents [10]. Many techniques have been proposed for the resolution of ISR such as intravascular brachytherapy, laser atherectomy, second stenting and drug-coated balloon (DCB) [11].

Balloon dilatation and stent implantation is associated with vascular injury, followed by repair processes that include endothelialization and neointimal formation, providing for the activation of the inflammatory response and the release of blood inflammatory markers [9,10,11]. The analysis of these biomarkers could prove useful to physicians to predict an eventual predisposition to ISR.

DCB has been shown to yield excellent results in the setting of PAD. Angiographic and clinical data highlight the superiority of this technique, and the European clinical practice guidelines endorsed its use in the treatment of ISR (class I, Evidence A) [11,12,13]. The DCB is a traditional drug-coated balloon, mainly with Paclitaxel, which significantly improves the short- and medium-term patency of angioplasty by minimizing neo-intimal hyperplasia [14,15,16,17]. Paclitaxel, with a nominal dose between 2 and 3.5 μg/mm^2^, has lipophilic and hydrophobic properties and, together with a hydrophilic agent, allows the drug to be released to the surface of the artery (Figure 1). There are more than 10 DCBs available in Europe, only three of which are FDA approved. It has been shown that the treatment of ISR with DCB produces better results than traditional angioplasty, while a superiority over drug-eluting stents has not yet been demonstrated [1,18,19]. DCBs are mainly used on the FP axis, particularly on the FP-ISR and rarely on the common iliac or femoral arteries. Another important topic that will be covered in this review are biomarkers. In the literature, microRNA is emerging with great strength as predictors or diagnostic elements for numerous pathologies. ISR, microRNA-320a, microRNA-3937, microRNA-642a-3p and microRNA-572 are proving to be excellent predictors both for evaluating the follow-up and for evaluating the longevity of the patency of the placed stent. This narrative review intends to summarize the value of circulating biomarkers as predictors of ISR, analyze the current knowledge on DCB on ISR and aims to identify the advantages of using DCB in the treatment of ISR of the FP district.

## 2. Materials and Methods

We checked the PubMed and Scopus databases from inception to March 2021. The following key word have been used, with relative MeSH terms: ((“drug eluting” OR “drug coated” OR “coated” OR “eluting” OR “paclitaxel”) AND (“balloon” OR “device” OR “devices” OR “endovascular” OR “stent”) AND (“peripheral” OR “femoral” OR “iliac” OR “popliteal” OR “tibial”) NOT “coronary”) AND (“review” OR “metanalysis” OR “meta-analysis”) AND (“in-stent restenosis”), or (((“drug eluting” OR “drug coated” OR “coated” OR “eluting” OR “paclitaxel”) AND (“balloon” OR “device” OR “devices” OR “endovascular” OR “stent”) AND (“peripheral” OR “femoral” OR “iliac” OR “popliteal” OR “tibial”) NOT “coronary”) AND (“in-stent restenosis”)) NOT (“review” OR “metanalysis” OR “meta-analysis”)). All the articles taken into consideration have been revised; even the references have been checked in order not to lose valuable information, and above all, to verify the relevance with the theme of this review. To be considered, the articles had to have studied ISR in PAD. We initially included randomized studies and observational non-randomized studies. Secondary research articles have been also investigated. We also included, in addition to systematic reviews, prospective studies that examined and compared IRS resolution techniques. Outcomes of interest for selected articles have been included in a shared dataset by three independent authors. All disagreements were resolved by consensus or after consultation with the senior author. Considering the narrative nature of this review, no statistical analysis has been performed.

## 3. Results

After preliminary evaluation, duplicate removal and manuscript screening, a total of 33 papers have been included in this review. The study design of the included studies is summarized in Table 1, while Table 2 shows the baseline characteristics of the included patients. In all the studies included in this review, the treatment with DCB demonstrated a higher rate of patency and minor rate of restenosis during the follow-up. Furthermore, a significant clinical improvement was highlighted in the majority of the studies, and this result acquires more value if we consider that these data have not been evaluated in all works. Table 3 outlines the results for each study and each treatment arm.

## 4. Diagnostic Implications

PTA of peripheral arteries is associated with vascular wall injury followed by repair processes, including endothelialization and neointimal formation [51]. This process is more expressed after stent implantation. The damage to the vascular wall caused by angioplasty and/or stenting causes inflammatory activation. This mechanism leads to the activation of the proliferative process, which consists of the proliferation, migration and differentiation of vascular smooth muscle cells (VSMC). Therefore, this inflammatory process leads to the release of various blood markers.

Beyond the classic inflammatory indices, microRNAs are assuming a considerable importance. Many scientific papers examine mRNAs as markers of disease or as markers of disease progression (Figure 2). In the case of ISR, we have noted and assumed from the literature that these markers can evaluate the longevity of a stent based on the degree of expression at the plasma level.

### 4.1. Inflammatory Markers

Numerous studies have reported the probable correlation between inflammatory markers and the restenosis process. In particular, some studies have previously reported that high levels of C-Reactive Protein (CRP) could predict the ISR after the BMS implantation [52,53]. Furthermore, Zhu et al., [54], conducted a meta-analysis of six prospective observational trials, which confirmed that the higher level of hs-CRP is associated with a significantly increased risk of ISR (OR 1.16, 95% CI 1.01–1.30; *p* < 0.05) in patients who underwent percutaneous transluminal coronary angioplasty. Finally, Jakubiak et al. [55], reported in a review that processes such as inflammation, neointimal hyperplasia and neoatherosclerosis, allergy, resistance to antimitotic drugs used for coating stents and balloons, genetic factors, and technical and mechanical factors, could be implicated in restenosis complications. The authors concluded that every effort should be made to develop knowledge about the pathogenesis of ISR after endovascular treatment of PAD, leading to the availability of more and more perfect therapeutic tools in clinical practice. Some studies reported that the activation of the complement system could play a crucial role in the pathogenesis of restenosis [56,57]. Speidl et al. [56] reported that a higher C5a plasma level is associated with an increased risk of restenosis in patients with PAD who underwent peripheral PTA. In this study C5a concentration was measured at baseline and eight hours after the procedure. Median C5a levels increased significantly from 39.7 ng/mL (IQR 27.8 to 55.0) at baseline to 53.8 ng/mL (IQR 35.6 to 85.1, *p* < 0.001) 8 h post intervention. During follow-up period, 53% of patients developed restenosis, and elevated levels of C5a at baseline were significantly associated with an increased risk for restenosis (*p* = 0.0092). Furthermore, the authors specified that this effect was independent of nonspecific inflammation as reflected by the plasma levels of CRP in their patients.

From these experiences, it comes to light that the inflammatory mechanisms play a major role in the development of restenosis after PTA and stent implantation. Therefore, the value of inflammatory biomarkers should be more investigated to improve patency rates.

### 4.2. microRNA

Recently, studies have reported that elevated levels of microRNA in patients after stenting could be predictive of ISR. Yuan et al. [58] reported that circulating microRNA-320a and microRNA-572 have promising value in diagnosing ISR in patients with PAD. The authors compared 78 patients with ISR, 68 non-ISR patients and 62 healthy volunteers. The microarray analysis showed significant changes in microRNAs, which were up-regulated or down-regulated in ISR groups compared with non-ISR and healthy volunteers. In fact, the analysis revealed that the expression of plasma microRNA-320a, microRNA-3937, microRNA-642a-3p and microRNA-572 were significantly higher in ISR patients than in the control groups. On the other hand, microRNA-4669 and microRNA-3138 showed significantly lower expression in the ISR group than that in the control groups. In addition, from the entire sample set, testing with quantitative reverse transcriptase-polymerase chain reaction (qRT-PCR) and receiver operating characteristic (ROC) analysis, the patients with ISR showed significantly higher expression levels of microRNA-320a and microRNA-572, suggesting their high potential diagnostic value for ISR detection.

Furthermore, Stojkovic et al. [59] recently reported a study on 62 consecutive PAD patients after infrainguinal PTA with stent implantation. The authors investigated the predictive value of 11 microRNAs for the composite endpoint of restenosis and atherothrombotic events (primary endpoint) and target lesion revascularization (TLR, secondary endpoint), demonstrating that the circulating microRNA-195 could predict restenosis, atherothrombotic events and TLR after PTA with stent implantation in FP district.

## 5. Discussion

The new treatments for PAD have considerably simplified the post-operative course compared to the open surgery approach, but a significant increase in ISR has been observed in recent years and is forecasted in the next years considering the number of procedures performed in each center [2,4,5,26,60,61]. Despite reducing the impact for the patient, PTA is giving very poor results in terms of long-term patency and TLR, and DCB is a new technique which is performing adequately in ISR.

As anticipated in the introduction, this manuscript has the intention of providing a narrative review on the current knowledge on the value of circulating biomarkers as predictors of ISR and to foster the scientific debate on the advantages of using DCB in the treatment of ISR in the FP district. In fact, the use of the stent in the FP district will become an increasingly pursued practice in the field of vascular surgery, making it increasingly necessary to study this knowledge in depth.

There is a focus on more complex and combined techniques for the treatment of ISRs which are producing more encouraging results such as DES, again with major limitations such as a stent length which excludes treating long stenosis. DCB shows superior results compared to traditional re-PTA with a stent and is becoming the technique of choice for ISRs [12,13,14,15,62]. Combined techniques such as laser atherectomy and DCB are also being developed and appear to be candidates to become the reference technique, but adequate studies are warranted [1,11,18,19]. In all the studies included in this review, the treatment with DCB demonstrated a higher rate of patency and minor rate of restenosis during the follow-up. Furthermore, Tepe et al. [24] in the COPACABANA trial reported that at the 12-month follow-up, TLR was performed in 18 (49%) of 37 patients in the uncoated group and in 6 (14%) of 43 patients in the single-dose DCB group (*p* = 0.001). At ~2 years after treatment, a remarkable number (14/27, 52%) of TLRs were recorded in the single-dose DCB group. The authors concluded that treatment with DCBs resulted in significantly less 6-month restenosis rate and fewer TLRs up to 24 months than the treatment with uncoated balloons.

In the recent metanalysis performed by Xi et al. [12], despite not considering the most recently published articles, 18 studies (9 RCTs and 9 OSs) have been included with a follow-up extended to 3 years. The data analysis showed how the treatment of ISR with DCB and DES was comparable, concluding that the treatment with DCB has a proven efficacy and certainly is not inferior to DES. The current guidelines confirm that DCB is a valid treatment for ISR, which obviates various problems brought about by the placement of traditional stents such as the reduction, in terms of time, of the double antiplatelet therapy, required in the placement of stent-in-stent. Another finding was that the restenosis has drastically decreased after the advent of DES and DCB. However, data have emerged that show that PAD patients treated with DCB (paclitaxel) have an increased risk of death, and it is hypothesized that paclitaxel toxicity is the cause of this increased risk. The mechanism of this increase remains unknown, and new studies are needed to investigate this issue. By comparing the DCBs used for the treatment of PAD and the DCBs used for coronary heart disease, it is hypothesized that the problem is the size and therefore the quantity of paclitaxel released. At the coronary level the DCB is very small, and consequently, the quantity of drug released is significantly lower than the DCB used for peripheral arteries (about 10 times greater). Another hypothesis, in addition to the intrinsic quantity of the drug present on the DCB, is the washing of the drug due to the blood circulation, obviously greater in the peripheral arteries than in the coronary circulation. Less circulating paclitaxel may be the reason for the diverging data between coronary and peripheral DCB. An alternative to PCB or Sirolimus-coated balloon (SCB) is being studied, which seems to give encouraging results. The mechanism of SCB is different from paclitaxel: the first falls into the class of cytostatics, the second into the class of cytotoxics, reversibly binding to the FKBP12 receptor forming a complex with rapamycin blocking the cell cycle in the G1 and S phase. Furthermore, Sirolimus has anti-inflammatory properties compared to paclitaxel, which appears to be the target treatment of patients with ISR. This new technology was approved by the European community and obtained the CE mark in 2016, and numerous centers are experimenting with this new method with Sirolimus. Sirolimus might be the alternative to paclitaxel in peripheral and coronary treatment, provided that the mortality data will be confirmed with scientific evidence.

In addition to all the data collected on the efficacy of the specific drug, the anatomy of the stenosis should also be carefully considered. Feng et al. [14] specifically evaluated the length of the stenosis by dividing the patients into two main categories. CTO (chronic total occlusion) >10 cm, in the reported analysis. Those treated with DCB had a 1-year patency free from other treatments, significantly lower compared to chronic total occlusion, with a length less than 10 cm. The measure of stenosis thus became a 1-year predictor of patency. In addition to the length of the stenosis, the degree of calcification was taken into consideration, considered a limiting factor for treatment with DCB13. It was noted that calcification in the arterial vessel reduces the diffusion capacity of the cytostatic/cytotoxic drug to the arterial wall. Another problem for which we are turning to combined procedures, as described above, is the degree of stenosis or calcification that does not allow for the passage of devices. For this reason, many stenoses or ISRs must be pretreated with traditional balloons to fragment the stenosing plaque, bringing the vessel back to an adequate caliber and allowing the absorption of drugs in situ [12,13,14,15].

## 6. Conclusions

ISR has been increasingly acknowledged as a daunting complication after percutaneous treatment of PAD. Poor long-term patency and the risk of stent failure in the case of inadequate pharmacological management expose the patient to a risk of limb injury. Multiple studies have shown that proinflammatory markers and high plasma levels of micro-RNA can influence the outcomes after peripheral stenting. This aspect could hold promise for early recognition of ISR, paving the way for future therapeutic tools. The DCB device has been recently introduced to perform target vessel and TLR. DCB have been shown to have promising results and will rapidly be the first line indication for ISR. In particular, DCB showed a significantly lower rate of restenosis and TLR in all the analyzed reports compared to conventional PTA. However, studies with dedicated long-term follow-up are warranted to verify the true efficiency of a technique. DCB gives excellent results in the short-term outcomes, and therefore it might be reliably speculated that it will become the best practice in complex and complicated PAD treatment.

## Figures and Tables

**Figure 1 diagnostics-12-02207-f001:**
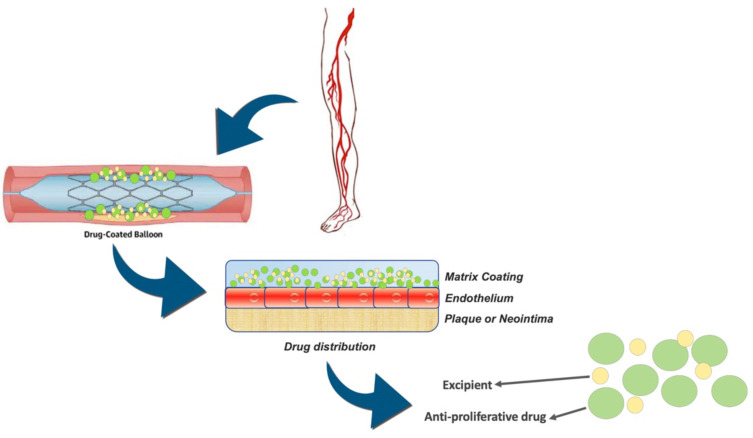
Mechanism of action of the drug-coated balloon and drug distribution through the endothelium.

**Figure 2 diagnostics-12-02207-f002:**
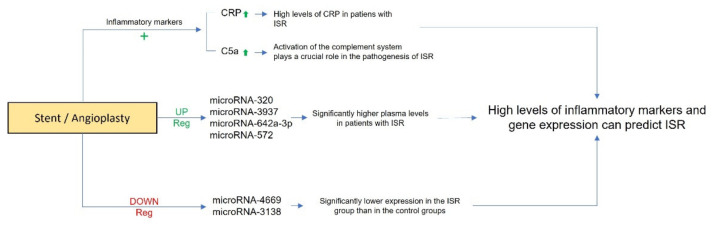
Schematic representation of diagnostic implications of inflammatory markers and microRNAs in in-stent restenosis. “+” indicates an increase.

**Table 1 diagnostics-12-02207-t001:** Study design of the included studies.

Study Name	Year	Study Design	Interventions	Inclusion Criteria	Exclusion Criteria	DCB Device	Reference
**Feng**	2021	Retrospective	DCB + DA	Claudication (Rutherford classes 2 and 3) or critical limb ischemia (Rutherford classes 4–6); femoropopliteal artery disease	Limited life expectancy (<2 years), acute thrombosis in the target vessel, prior bypass surgery, or thrombolysis within the past 6 weeks	DCB: Orchid; Acotec Scientific, Beijing, China; DA: SilverHawk or TurboHawk devices (Medtronic, formerly Covidien/ev3, Plymouth, MN, USA)	[14]
**Kokkinidis**	2020	Retrospective	DCB + LA vs. PTA + LA	Tosaka II or Tosaka III- FP-ISR	Not available	DCB = IN.PACT Admiral paclitaxel-eluting balloon [Medtronic Cardiovascular, Santa Rosa, CA] or Lutonix [Bard Peripheral Vascular, Tempe, AZ]; LA = Turbo Elite (Philips Inc, Colorado Springs, CO), the Turbo-Power excimer laser (Philips Inc), and the Turbo Tandem device (Philips Inc)	[18]
**Kokkinidis et al.**	2018	Retrospective	Laser + DCB/Laser + PTA	Patients (mean age 70.3 ± 10.6 years; 86 men) with Tosaka II (n = 29) and Tosaka III (n = 83) FP-ISR retrieved from the vascular databases of 2 academic centers	Not available	Laser Atherectomy + DCB Laser Atherectomy + PTA	[19]
**PATENT/Schmidt et al.**	2014	Multicentre RCT	Laser + PTA	Patients between 18 and 85 years of age with peripheral artery disease ranging from intermittent claudication to CLI (Rutherford categories 2 to 5), an existing femoropopliteal stent with suspected ISR, and an ankle-brachial index (ABI) of 0.8 in the target limb. Anatomical inclusion criteria included angiographic evidence of significant ISR 50% stenosis by visual assessment within a previously deployed femoropopliteal stent(s). The target lesion had to be 1 cm and 25 cm, with a reference vessel diameter of 3.5 mm and 7 mm. A minimum of one patent tibial vessel to the ankle without stenosis requiring intervention prior to the 1-month follow-up was required. Additional anatomical inclusion criteria included a popliteal artery free of visually assessed stenosis 0.50% within 1 cm of the knee joint in the anteroposterior projection and angiographic documentation of successful intraluminal guidewire crossing.	Major exclusion criteria included a life expectancy of 12 months, treatment to the target lesion of 3 months prior to study enrollment, serum creatinine of 2.5 mg/dL unless dialysis-dependent, 0.50% diameter stenosis following treatment of inflow lesions in the iliac or common femoral artery, aneurysm within the target lesion, or grade 4/5 stent fracture affecting the target stent or proximal to the target stent	turbo-booster and excimer laser	[20]
**Brodmann et al.**	2017	Single arm prospective	DCB	Patients enrolled at sites qualified by the VasCore Duplex Core Lab (Boston, Massachusetts) were screened to meet one or more of the imaging criteria based on an algorithm. The hierarchy for the imaging cohort subgroup assignment was as follows: (1) de novo ISR; (2) long lesions $15 cm; and (3) chronic total occlusions $5 cm. Enrollment in the respective imaging cohorts was only open to subjects that had a de novo ISR, long lesion, or chronic total occlusion at pre-procedure baseline. The only subjects who were included in the respective imaging cohort analyses, however, were those who had the primary target lesions that met the criteria of the respective cohort during the index procedure (i.e., no other types of lesions could have been treated during the index procedure). Subjects with symptoms of intermittent claudication or ischemic rest pain (Rutherford clinical category 2 to 4) and angiographic evidence of occlusion or stenosis (length $2 cm) in the SFA or popliteal artery (including P1 to P3 segments) were eligible for enrollment in the IN.PACT Global study	Subjects with multiple lesions were allowed. Subjects with tissue loss were excluded.	INPACT Admiral	[21]
**Horie**	2021	Retrospective	PEB vs. PTA	Age >50 years, symptomatic PAD (Rutherford category 2 to 5), ISR > 70% at the stented site in femoropopliteal segments	Acute limb ischemia and/or short life expectancy	IN.PACT Pacific PCB (Medtronic, Minneapolis, MN)	[22]
**DEBATE-ISR/Liistro F**	2014	Multicentre RCT	DCB/PTA	Diabetic patients with femoropopliteal ISR	Paclitaxel allergy; contraindication to combined antiplatelet treatment; life expectancy <1 year	DCB = In Pact Admiral (Medtronic) PTA = Unspecified	[23]
**COPA CABANA/Tepe G et al.**	2020	Multicenter RCT	DCB vs. ba	ISR ≥70% or in-stent occlusion 3 to 27 cm long within the stent and adjacent segments of the SFA and/or popliteal artery occurring >3 months after stent implantation; (2) Rutherford category 2 to 5 ischemia; (3) at least 1 patent runoff vessel; and (4) patient willingness and ability to continue study participation after the initial study procedure.	(1) No patent distal runoff vessel; (2) guidewire unable to cross the lesion or a planned sub-intimal approach to the ISR lesion; (3) presence of stent fracture grades 2 to 4; (4) persistent inflow lesion, acute thrombosis of the study lesion or planned major amputation (above the ankle); (5) aneurysm in the target vessel; (6) platelet count <100,000/mm^3^ or >700,000/mm^3^, leukocyte count <3000/mm^3^; (7) contraindication to paclitaxel or any anticoagulation or antiplatelet agent (eg, aspirin, heparin, clopidogrel, ticlopidine, abciximab); and (8) known intolerance or contraindications to contrast agents	Cotavance paclitaxel- coated balloon (MEDRAD Inc, Warrendale, PA, USA)	[24]
**BIOLUX P-III/Tepe G. et al.**	2021	Prospective non randomized trial	POBA/CB/SB/AD + DCB (Passeo-18 1x)	Patients were eligible if they had lesions in the infrainguinal arteries suitable for endovascular treatment with the Passeo-18 1x DCB, including below-the-knee lesions and Rutherford 5 or 6, under real-world conditions.	Excluded were patients with a life expectancy of less than one year, or failure to successfully cross the target lesion with a guide wire	Passeo-18 1x (DCB (BIOTRONIK AG, Switzerland)	[25]
**Liao CJ et al.**	2019	Retrospective	DCB/PTA	Patients with femoropopliteal occlusive disease treated with DA.	Yntreated ipsilateral iliac artery stenosis, ongoing dialysis treatment, aneurysm within target lesion, known intolerance or allergy to aspirin, heparin, clopidogrel, paclitaxel, or contrast agent, planned amputation of the target limb; and life expectancy <1 year.	SilverHawk	[26,27]
**ISAR-PEBIS/Ott I et al.**	2017	Multicentre RCT	DCB/PTA	Symptomatic ISR >70% or occlusion of SFA	Acute ischemia, thrombosis, untreated ipsilateral iliac artery stenosis >70%, severe renal insufficiency, life expectancy <1 year, contraindication to study medications	DCB = In Pact Admiral (Medtronic) PTA = Pacific Xtreme (Medtronic)	[28]
**FAIR/Krankenberg H et al.**	2015	Multicentre RCT	DCB/PTA	A SFA ISR up to 20 cm, stenosis >70%; one nfrapopliteal for distal runoff; Rutherford category 2–4.	An untreated ipsilateral iliac artery stenosis; ongoing dialysis treatment; treatment with oral anticoagulants	DCB = In Pact (TM) Admiral (Medtronic) PTA = Admiral Xtreme (Medtronic)	[29]
**PACUBA/Kinstner CM et al.**	2016	Multicentre RCT	DCB/PTA	Age >50 years, symptomatic PAD, ISR >50% in the SFA and P1 segment of the popliteal artery, at least 1 patent tibial vessel with distal runoff, Rutherford category 2–3	Inability to write informed consent; contraindication to study medications; and creatinine >2.5 mg/dL	DCB = Freeway 0.035 DCB (Eurocor) PTA = Unspecified	[30]
**Armstrong et al.**	2015	Retrospective	Laser + PTA/PTA	Symptomatic patients (mean age 71 years; 76 men) who underwent endovascular treatment of femoropopliteal ISR between 2006 and 2013.	Not available	Laser Catheter (Turbo Elite)	[31]
**SALVAGE/Laird et al.**	2012	Multicentre RCT	Laser + PTA	Patients between 18 and 90 years of age with either moderate-to-severe intermittent claudication or critical limb ischemia (Rutherford categories 2–5). Noninvasive lower extremity arterial studies (resting or exercise) in these subjects had to demonstrate an ankle-brachial index (ABI) equal to or less than 0.8 in the affected leg. Anatomic inclusion criteria included angiographic evidence of significant restenosis (defined as =50% by visual estimate) within a previously deployed femoropopliteal nitinol stent; target lesion length of ISR = 4 cm; femoropopliteal reference luminal diameter of =4.8 mm; and a minimum of one vessel infrapopliteal run-off defined as a one patent tibial vessel to the ankle with less than 50% stenosis.	Patients were excluded if they had a life expectancy <12 months; had undergone previous treatment to the target limb within 3 months of the study procedure; if the target lesion is within or adjacent to an aneurysm; if there were inflow-limiting lesions untreatable in this procedural setting; outflow lesions with >50% stenoses that require treatment in this procedural setting or within the 30-day follow-up; if a trans-lesional gradient >15 mm Hg persisted after optimal laser debulking and PTA; if angiographic evidence of intra-arterial thrombus or atheroembolism from inflow treatment was present; or if there was Grade 4 or 5 stent fracture in the restenotic stent	Excimer Laser with Adjunctive Balloon Angioplasty and Heparin-Coated Self-Expanding Stent Grafts	[32]
**Gandini et al.**	2013	RCT	Laser + DCB/DCB	Patients with chronic (0.30 days old) SFA in-stent reste.nosis, with no angiographically detectable antegrade blood flow and recurrent symptoms, were eligible for enrollment if they were 0.18 years old and poor candidates for surgical bypass due to comorbidities or anatomical restraints	Acute leg ischemia with sudden symptom onset and indication for thrombolytic therapy, poor distal SFA outflow (1 vessel), and pregnancy	Laser Atherectomy and Drug-Eluting Balloon Angioplasty	[33]
**Shammas et al.**	2016	RCT	JetStream XC + PTA	≥50% in-stent restenotic lesion in the superficial femoral or popliteal arteries (estimated diameter ≥5 mm), Rutherford category 1–5 ischemia, and at least 1 patent infrapopliteal runoff vessel.	Patients were excluded if they were not able to give informed consent, had a creatinine level >2.5 mg/dL, were unable to take antiplatelet drugs, or had a planned surgical or endovascular procedure within 15 days of the index procedure.	JetStream Rotational and Aspiration Atherectomy	[34]
**Sixt et al.**	2013	RS	SilverHawk + DCB/SilverHawk + PTA	Patients with femoropopliteal occlusive disease treated with DA. Rutherford classification (2 to 5)	not available	SilverHawk (LS and LX)	[35]
**PLAISIR/Bague N et al.**	2017	Non randomized observational study	DCB	Age 18 years old Symptomatic patient according to Rutherford Class 1, 2, 3, 4 or 5 Clinical degradation by at least 1 Rutherford stage or absence of healing of all skin lesions Symptoms related to SFA ISR defined by PSVR >2.4 within 3 e24 months after SFA stenting of de novo atherosclerotic lesions. Each patient may have either one or both limbs treated in the study The target ISR lesion is fully contained between the origin of the SFA and the distally femoropopliteal crossover (crossing by SFA of medial rim of femur in the PA projection) Adequate SFA inflow and outflow either pre-existing or successfully re-established (outflow defined as patency of at least one infragenicular artery) The target lesion must not extend beyond the stent margin Successful crossing of the target lesion, inflow and outflow lesions with a guidewire; Patient belongs to the French health care system	No atheromatous disease, Asymptomatic lesion. Known allergies to heparin, aspirin, other anti-coagulant/antiplatelet therapies, and/or paclitaxel Acute limb ischaemia; Patient on oral anticoagulation therapy Target lesion requires/has been pretreated with alternative therapy such as: DES, laser, atherectomy, cryoplasty, cutting/scoring balloon, etc; Life expectancy <1 year; Patient involved in another trial; Refusing patient Pregnancy; Patients receiving anticoagulation	INPACT Admiral	[36]
**Stabile et al.**	2012	Non randomized observational study	DCB	Superficial Femoral Artery In-Stent Restenosis	Not available	INPACT Admiral	[37]
**Virga et al.**	2014	Non randomized observational study	DCB	Superficial Femoral Artery In-Stent Restenosis	Not available	INPACT Admiral	[38]
**Milnerowicz et al.**	2019	Non randomized observational study	RA + DCB	Symptomatic patients (mean age 66.7 ± 9.7 years; 49 men) with total occlusion of a previously implanted stent	Not available	Elutax	[39]
**Giannopoulos et al.**	2020	Retrospective	DCB + LA vs. PTA + LA	Tosaka II or Tosaka III	Not available	DCB: InPact; LA: Turbo-PowerTM laser (Spectranetics Inc, Colorado Springs, CO, USA), Turbo-EliteTM (Spectranetics Inc) and Turbo-TandemTM (Spectranetics Inc)	[40]
**Thieme et al.**	2017	Non randomized observational study	DCB	(1) Patients age 18 years or older; (2) Rutherford Classification category of <4; (3) stenotic or obstructive vascular lesions of the femoropopliteal arteries; (4) lesions treatable with available Lutonix 035 DCB, size per current European Instructions for use version 6 (IFU); (5) at least one patent native outflow artery to the ankle free from significant lesion ($50% stenosis) as confirmed by angiography; and (6) informed consent and willingness to comply with the follow-up schedule.	(1) Enrolled in another clinical trial; (2) unable to take recommended medications as stated in IFU or had a noncontrollable allergy to contrast; (3) pregnant or planning on becoming pregnant; (4) intending to father a child; and (5) Rutherford category >4	Lutonix	[41]
**Shammeri et al.**	2012	Retrospective	VSG	femoropopliteal in-stent restenosis	not available	Viabahn Stent-Graft	[42]
**RELINE/Bosiers et al.**	2015	Multicentre RCT	VSG/PTA	Patient presents with lifestyle-limiting claudication, rest pain, or minor tissue loss (Rutherford category 2–5) Patient is willing to comply with specified follow-up evaluations at the specified times Patient is >18 years old Patient understands the nature of the procedure and provides written informed consent prior to enrollment in the study Patient has a projected life expectancy of at least 24 months Noninvasive lower extremity arterial studies (resting or exercise) demonstrate ankle-brachial index ≤0.8 Patient is eligible for treatment with the Viabahn Endoprosthesis Male, infertile female, or female of child bearing potential practicing an acceptable method of birth control with a negative pregnancy test within 7 days prior to study procedure Angiographic Restenotic or reoccluded lesion located in a stent that was previously implanted (>30 days) in the superficial femoral artery (SFA), suitable for endovascular therapy Total target lesion length between 4 and 27 cm (comprising in-stent restenosis and adjacent stenotic disease) Minimum of 1.0 cm of healthy vessel (non-stenotic) both proximal and distal to the treatment Popliteal artery patent at the intercondylar fossa of the femur to P3 Target vessel diameter visually estimated to be >4 and <7.6 mm at the proximal and distal treatment segments within the SFA Guidewire and delivery system successfully traversed lesion Angiographic evidence of at least 1-vessel runoff to the foot that does not require intervention (<50% stenotic)	Untreated flow-limiting aortoiliac stenotic disease Presence of a chronic total occlusion, i.e., a complete occlusion of the failed bare stent that cannot be reopened with thrombolysis or does not allow easy passage of the guidewire Any previous surgery in the target vessel Severe ipsilateral common/deep femoral disease requiring surgical reintervention. Perioperative unsuccessful ipsilateral percutaneous vascular procedure to treat inflow disease just prior to enrollment. Femoral or popliteal aneurysm located at the target vessel. Nonatherosclerotic disease resulting in occlusion (eg, embolism, Buerger’s disease, vasculitis) No patent tibial arteries (>50% stenosis). Prior ipsilateral femoral artery bypass. Severe medical comorbidities (untreated coronary artery disease/congestive heart failure, severe chronic obstructive pulmonary disease, metastatic malignancy, dementia, etc.) or other medical condition that would preclude compliance with the study protocol or result in a 2-year life expectancy. Serum creatinine >2.5 mg/dL within 45 days prior to study procedure unless the subject is currently on dialysis. Major distal amputation (above the transmetatarsal) in the study or nonstudy limb Septicemia or bacteremia. Any previously known coagulation disorder, including hypercoagulability Contraindication to anticoagulation or antiplatelet therapy. Known allergies to stent or stent-graft components (nickel-titanium or polytetrafluoroethylene). Known allergy to contrast media that cannot be adequately pre-medicated prior to the study procedure. Patient with known hypersensitivity to heparin, including those patients who have had a previous incidence of heparin-induced thrombocytopenia type II. Currently participating in another clinical research trial, unless approved by W.L. Gore & Associates in advance of study enrollment. Angiographic evidence of intra-arterial thrombus or atheroembolism from inflow treatment. Any planned surgical intervention/procedure within 30 days of the study procedure. Target lesion access not performed by transfemoral approach.	Viabahn Stent-Graft	[43]
**ZILVER PTX/Zeller et al.**	2013	Multicentre RCT	DES	(1) De novo or restenotic lesions of the above-the-knee segment of the femoropopliteal artery with 50% diameter stenosis and baseline clinical symptoms classified as Rutherford category (2) Patients could have multiple lesions requiring treatment, a history of prior stent placement within the lesion, bilateral lesions requiring treatment, and lesions of unlimited length.	Patients treated for multiple lesion types (e.g., both de novo and ISR) were excluded from this analysis	Zilver PTX drug eluting stents	[44]
**Murato et al.**	2016	Retrospective	DES/PTA	Femoropopliteal In-Stent Restenosis	not available	Zilver PTX drug eluting stents	[45]
**Van Den Berg et al.**	2014	Single arm prospective	Laser + DCB	Clinically relevant (Rutherford 3–6) ISR who were treated with excimer-laser angioplasty and drug-eluting balloons + clinical follow-up of at least 9 months	Not available	INPACT Admiral	[46]
**PERMIT-ISR Trial**	2021	Prospective non randomized trial	MATH + DCB	FP artery ISR defined as a peak systolic velocity (PSV) ratio > 2.0 (PSV ≥ 200–250 cm/s) at the target lesion (9); angiographic evidence of significant ISR > 70% by visual assessment within the stent; Rutherford category 2 to 6; reference vessel diameter of 4–8 mm; target ISR lesion >1 and <35 cm; at least 1 non-occluded crural vessel runoff; and ankle–brachial index (ABI) < 0.6.	Serious renal failure (serum creatinine > 2.5 mg/dL), treatment of the target lesion within 3 months before study enrollment, aneurysm in the target lesion, stent fracture, planned amputation of the target limb, and expected follow-up time < 2 years.	MATH: RotareR S System (Straub Medical, Wangs, Switzerland); DCB = OrchidR paclitaxel-coated balloons (Orchid, AcoTec, Beijing, China)	[47]
**Tomoi Y. et al.**	2021	Non randomized observational study	POBA and DES vs. BP	Elective treatment of femoropopliteal (FP) in-stent occlusion	Urgent setting, elective treatment with POBA and BMS implantation	not available	[48]
**Zhang B. et al.**	2020	Non randomized observational study	RA/RT + POBA + DCB	Patients with femoropopliteal Tosaka class III ISR lesions treated with DCB from September 2016 to September 2018 were enrolled in this single-center study. The inclusion criteria were as follows: (1) patients age 18 years or older; (2) patients diagnosed with Rutherford Classification category 1 or greater; (3) the presence of femoropopliteal Tosaka III ISR lesions; and (4) the lesions were treatable with the available Acotec Orchid DCB.	Pregnant patients or patients who were planning on becoming pregnant were excluded.	Orchid DCB (Acotec, Beijing, China)	[49]
**Bosiers M. et al.**	2020	Multicenter RCT	POBA + VSG vs. POBA	General inclusion criteria (1). Patient presenting with lifestyle-limiting claudication, rest pain or minor tissue loss (Rutherford classification from 2 to 5) (2). Patient is willing to comply with specified follow-up evaluations at the specified times (3). Patient is >18 years old (4). Patient understands the nature of the procedure and provides written informed consent, prior to enrolment in the study (5). Patient has a projected life-expectancy of at least 24 months (6). Noninvasive lower extremity arterial studies (resting or exercise) demonstrate ankle-brachial index ≤0.8 (7). Patient is eligible for treatment with the Viabahn^®^ Endoprosthesis (W.L. Gore) (8). Male, infertile female or female of child bearing potential practicing an acceptable method of birth control with a negative pregnancy test within 7 days prior to study procedure Angiographic inclusion criteria (1). Restenotic or reoccluded lesion located in a stent which was previously implanted (>30 days) in the superficial femoral artery, suitable for endovascular therapy (2). Total target lesion length between 4 and 27 cm (comprising in-stent restenosis and adjacent stenotic disease) (3). Minimum of 1.0 cm of healthy vessel (non-stenotic) both proximal and distal to the treatment area (4). Popliteal artery is patent at the intercondylar fossa of the femur to P 3 (5). Target vessel diameter visually estimated to be >4 mm and <7.6 mm at the proximal and distal treatment segments within the SFA (6). Guidewire and delivery system successfully traversed lesion (7). There is angiographic evidence of at least one-vessel-runoff to the foot, that does not require intervention (<50% stenotic)	(1). Untreated flow-limiting aortoiliac stenotic disease (2). Presence of a chronic total occlusion, i.e., a complete occlusion of the failed bare stent that cannot be re-opened with thrombolysis or does not allow easy passage of the guidewire by the physician (3). any previous surgery in the target vessel (4). severe ipsilateral common/deep femoral disease requiring surgical reintervention (5). Perioperative unsuccessful ipsilateral percutaneous vascular procedure to treat inflow disease just prior to enrollment (6). Femoral or popliteal aneurysm located at the target vessel (7). Non-atherosclerotic disease resulting in occlusion (e.g., embolism, Buerger’s disease, vasculitis) (8). No patent tibial arteries (>50% stenosis) (9). Prior ipsilateral femoral artery bypass (10). Severe medical comorbidities (untreated CAD/CHF, severe COPD, metastatic malignancy, dementia, etc.) or other medical condition that would preclude compliance with the study protocol or 2-year life expectancy (11). serum creatinine >2.5 mg/dl within 45 days prior to study procedure unless the subject is currently on dialysis (12). Major distal amputation (above the transmetatarsal) in the study or non-study limb (13). Septicemia or bacteremia (14). Any previously known coagulation disorder, including hypercoagulability (15). Contraindication to anticoagulation or antiplatelet therapy (16). Known allergies to stent or stent graft components (nickel-titanium or ePTFE) (17). Known allergy to contrast media that cannot be adequately premedicated prior to the study procedure (18). Patient with known hypersensitivity to heparin, including those patients who have had a previous incidence of heparin-induced thrombocytopenia (HIT) type II (19). Currently participating in another clinical research trial, unless approved by W.L. Gore & Associates in advance of study enrolmen t (20). Angiographic evidence of intra-arterial thrombus or atheroembolism from inflow treatment (21). Any planned surgical intervention/procedure within 30 days of the study procedure (22). Target lesion access not performed by transfemoral approach	Viabahn Stent-Graft	[50]

RCT = randomized control trial, PSMs = propensity score matched study, MPRT = multicenter prospective randomized controlled trial, NRSO = not randomized observational studies, PEB = paclitaxel eluting balloon, EES = everolimus eluting stent, PES = paclitaxel eluting stent, DES = drug eluting stent, DCB = drug coated balloon, BMS = bare metal stent, LD = laser debulking, PTA = percutaneous transluminal angioplasty, POBA = plain old balloon angioplasty, LA = laser atherectomy, RA = rotational atherectomy, VSG = Viabahn Stent-Graft, CSD = covered stend deployment, RS = retrospective study, PMT = prospective multicenter trial, DCBA = drug coated balloon angioplasty, DA = directional atherectomy, MATH = percutaneous mechanical atherectomy plus thrombectomy.

**Table 2 diagnostics-12-02207-t002:** Summary of baseline characteristics.

Study Name	Patients	Mean Age	Diabetes %	Rutherford Class	Duration od Follow-up (m)	Drug Treatment	Reference
**Feng**	79	70.9	72.2	35 (>4)	12	DAPT	[14]
**Kokkinidis**	117	70.0 ± 11.0	DCB + LA = 33/66; BA + LA = 27/51	DCB + LA = 27 (>4); BA + LA = 12 (>4)	24	DAPT	[18]
**Kokkinidis et al.**	LD + DCB = 62	68.5 ± 10	50.08	-	12	ASA = 49	[19]
	LD + PTA = 50	72.5 ± 10.8	52	-	12	ASA = 44	
**Schmidt et al.**	Laser + PTA = 90	69.5 ± 9.3	50	6 (>4)	12	-	[20]
**Brodmann et al.**	DCB = 131	67.8 ± 10.1	35.1	87 (>3)	12	-	[21]
**Horie**	50	PEB = 71.9 ± 7.5; BA = 71.5 ± 8.9	PEB = 76.0; BA = 68.0	0 (>4); PEB = 24/25 (2/3); BA = 25/25 (2/3)	60 (PEB = 19/25; BA = 20/25)	DAPT	[22]
**DEBATE-ISR/Liistro F**	44	32	100	33 (>4)	36	DAPT	[23]
**COPA CABANA/Tepe G et al.**	47	68.3 ± 9.6	42.5	3 (>4)	22	DAPT	[24]
**BIOLUX P-III/Tepe et al.**	DCB = 103	70.4 ± 9.8	42.7	75.3% (≥3)	24	/	[25]
**Liao CJ et al.**	74 patients (DCB n = 38)/PTA (n = 36))	66.8 ± 7.9	DCB 50.0/ PTA 47.2	2 18.4 vs. 22.2	12	clopidogrel + ASA 3 days before and for 3 months; aspirin was continued as a permanent therapy	[26,27]
				3 36.8 vs. 41. 7			
				4 39.5 vs. 30.5			
				5 5.3 vs. 5.6			
**ISAR-PEBIS/Ott I et al.**	36	70 ± 10	4.32	1 (>4)	24	DAPT	[28]
**FAIR/Krankenberg H et al.**	62	69 ± 8	17.36	3 (>4)	12	DAPT	[29]
**PACUBA/Kinstner CM et al.**	35	68.1 ± 9.2	48.5	0 (>4)	12	DAPT	[30]
**Armstrong et al.**	LD +PTA = 54	73 ± 11	55.5	19 (>4)	24	ASA (47) Plavix (33)	[31]
	PTA = 81	69 ± 11	55.5	37 (>4)	24	ASA (77) Plavix (60)	
**SALVAGE/Laird et al.**	Laser + PTA = 27	70.0 ± 10.5	59.2	2 (>4)	12	-	[32]
**Gandini et al.**	Laser + DCB = 24	74.1 ± 7.2	-	-	12	-	[33]
	DCB = 24	70.1 ± 11.6	-	-	12	-	
**Shammas et al.**	JetStream XC + PTA = 29	69.9 ± 11.7	41	10 (>4(	6	ASA = 29	[34]
**Sixt et al.**	SilverHawk + DCB = 29	70 ± 13	48	11 (>4)	12	ASA (27) Plavix (28)	[35]
	SilverHawk + PTA = 60	68 ± 10	43	10 (>4)	12	ASA (54) Plavix (56)	
**PLAISIR/Bague N et al.**	DCB = 53	69 ± 12	30.18		18	53	[36]
**Stabile et al.**	DCB = 39	65.9 ± 9.6	48.7	2.9 ± 0.7	12	ASA (39)	[37]
**Virga et al.**	DCB = 39	65.9 ± 9.6	48.7	2.9 ± 0.7	24	ASA (39)	[38]
**Milnerowicz et al.**	DCB = 74	66.7 ± 9.7	25	37 (>4)	12	ASA (74)	[39]
**Giannopoulos et al.**	Turbo Power + DCB = 27	64,7	59	27 (>3)	12	ASA (19) Plavix (7)	[40]
	Laser + BA = 51	72,5	52.9	46 (>3)	12	ASA (45) Plavix (36)	
**Thieme et al.**	DCB = 89	68.2 ± 9.65	28.1	8 (>4)	24	-	[41]
**Shammeri et al.**	DES = 26	73	55	28 (>3)	36	ASA (26) Plavix (26)	[42]
**RELINE/Bosiers et al.**	DES = 39	67.7 ± 9.8	33.3	27 (>3)	12	-	[43]
	PTA = 44	69.0 ± 9.7	36.4	38 (>3)	12	-	
**ZILVER PTX/Zeller et al.**	DES = 108	68.3 ± 9.4	38.9	-	12	Plavix = 108	[44]
**Murato et al.**	DES = 57	74 ± 9	60	-	12	DAPT = 57	[45]
	PTA = 44	69 ± 11	57	-	12	DAPT = 44	
**Van Den Berg et al.**	Laser + DCB	78 ± 6.5	-	-	9	-	[46]
**PERMIT-ISR Trial**	59	71.0 ± 11.2	45.8	48 (class 4–6)	33 ± 8	single or dual antiplatelet therapy	[47]
**Tomoi Y et al.**	DES = 28	71.2 ± 9.1	50	3 (3, 4)	36.6 ± 25.5	DAPT (2 m) then ASA	[48]
**Zhang B. et al.**	DCB = 28	69.3 ± 8.5	67.9	3 (60.7%)	21.5 ± 10.3	DAPT (6 m) then single	[49]
**Bosiers M. et al.**	VSG = 39	67.69 ± 9.77	33.3	3 (56.4%)	24	DAPT (6 m) then ASA	[50]

RCT = randomized control trial, PSMs = propensity score matched study, MPRT = multicenter prospective randomized controlled trial, NRSO = not randomized observational studies, PEB = paclitaxel eluting balloon, EES = everolimus eluting stent, PES = paclitaxel eluting stent, DES = drug eluting stent, DCB = drug coated balloon, BMS = bare metal stent, LD = laser debulking, PTA = percutaneous transluminal angioplasty, POBA = plain old balloon angioplasty, LA = laser atherectomy, RA = rotational atherectomy, VSG = Viabahn Stent-Graft, CSD = covered stend deployment, RS = retrospective study, PMT = prospective multicenter trial, DCBA = drug coated balloon angioplasty, DA = directional atherectomy, MATH = percutaneous mechanical atherectomy plus thrombectomy.

**Table 3 diagnostics-12-02207-t003:** Summary of study outcomes.

Study Name	Patients	Follow-Up Duration	Interventions	Patency	Restenosis	TLR	TVR	Stent Thrombosis	Amputations	Clinical Improvement	Reference
**Feng**	**79**	12 months	DCB + DA	80.8 1 yy		7.8	Superficial femoral and popliteal artery	0	1.2		[14]
**Kokkinidis et al.**	**62**	12 months	Laser + DCB	86.7	13.3	27.5	Femoropopliteal		5.2		[19]
	**50**	12 months	Laser + PTA	56.9	43.1	49.5	Femoropopliteal		2.6		
**Schmidt et al.**	**90**	12 months	Laser + PTA	37.8	62.2	27.4	Femoropopliteal	2.2	0	Improvement in the Rutherford-Becker categories. ABI, WIQ	[20]
**Brodmann et al.**	**131**	12 months	DCB	88.7	11,3	7.3	Femoropopliteal	0.8	0	75	[21]
**Horie**	**50**	18 months	PEB = 25; PTA = 25	PEB = 65.7; PTA = 18.7	PEB = 24.0; PTA = 80.0	PEB = 16.0; PTA = 44.0	Superficial femoral and proximal popliteal artery	0	0		[22]
	**117**	24 months	DCB + LA = 66; PTA + LA = 51	DCB + LA = 45; PTA + LA = 24	DCB + LA = 55; BA + LA = 76	DCB + LA = 49; PTA + LA = 55	Superficial femoral and proximal popliteal artery	0	0		
**DEBATE-ISR/Liistro F**	**44**	12 months	DCB	80.50	19.5	13.6	Superficial femoral proximal popliteal		0	77.3	[23]
	**42**	12 months	PTA	28.20	71.8	31	Superficial femoral proximal popliteal		2.4	59.5	
**COPA CABANA/Tepe G et al.**	**47**	12 months	DCB	86.00	14, 6 m	14, 12 m	Superficial femoral proximal popliteal		4.2	24	[24]
	**41**	12 months	PTA	41.00	59, 6 m	49, 12 m	Superficial femoral proximal popliteal		2.4	15	
**Tepe G. et al.**	**103**	24 months	POBA/CB/SB/AD + DCB (Passeo-18 1x)	77.3, 12 m/58.5, 24 m		10.8, 12 m/21.6, 24 m	Infrainguinal arteries		Major 0; Minor 3.2 at 12 m 4.5 at 24 m		[25]
**Liao CJ et al.**	**38**	12 months	DCB	89.5	10.5	6.1	Femoropopliteal		0	75.8	[26,27]
	**36**	12 months	PTA	58.3	41.7	35.5			0	51.6	
**ISAR-PEBIS/Ott I et al.**	**36**	24 monts	DCB	70, 6 m	30, 6 m	19, 24 m	Superficial femoral	3, 24 m	0, 24 m		[28]
	**34**	24 months	PTA	41, 6 m	59, 6 m	50, 24 m	Superficial femoral	0, 24 m	0, 24 m		
**FAIR/Krankenberg H et al.**	**62**	12 months	DCB	70.50	29.5	9.2	Superficial femoral	2.1	0	77.8	[29]
	**57**	12 months	PTA	37.50	62.5	47.4	Superficial femoral	4.5	0	52.3	
**PACUBA/Kinstner CM et al.**	**35**	12 months	DCB	40.70	59.3	51	Superficial femoral proximal popliteal	2.8	0	68.8	[30]
	**39**	12 months	PTA	13.40	86.6	77.9	Superficial femoral proximal popliteal	0	0	54.5	
**Armstrong et al.**	**54**	24 months	Laser + PTA		I/II: 6 9 III: 69, 24 m	class I/II FP-ISR:14 III: 43, 24 m	Femoropopliteal	class I/II FP-ISR:26 III: 12, 24 m	0	89, 1 m	[31]
	**81**	24 months	PTA		I/II: 46 III: 100%, 24 m	class I/II FP-ISR: 44 III: 48, 24 m	Femoropopliteal	class I/II FP-ISR:33 III: 71, 24 m	0	81, 1 m	
**SALVAGE/Laird et al.**	**27**	12 months	Laser + PTA	48	52	17.4	Femoropopliteal	0	0	improvements in ABI, walking distance, walking speed, and ability to climb stairs	[32]
**Gandini et al.**	**24**	12 months	Laser + DCB	66.7	33.3	16.7	Superficial femoral		8	Limb salvage 91.7 Healing ulcer 87.5	[33]
	**24**	12 months	DCB	37.5	62.5	50	Superficial femoral		46	Limb salvage 54.2 Healing ulcer 62.5	
**Shammas et al.**	**29**	12 months	JetStream XC + PTA	72, 6 m	28	14, 6 m/41, 12 m	Femoropopliteal	0	0	improved significantly	[34]
**Sixt et al.**	**29**	12 months	SilverHawk + DCB/	84.7	15.3		Femoropopliteal		2/89 all enrolled	no difference between the treatment groups regarding improvement in clinical status	[35]
	**60**	12 months	SilverHawk + PTA	43.8	56.2				2/89 all enrolled		
**PLAISIR/Bague N et al.**	**53**	18 months	DCB	78.1	21.9	23.4 18 m	Femoropopliteal	9	1.88	67 18 m	[36]
**Stabile et al.**	**39**	12 months	DCB	100	0		Superficial femoral	0	0	100	[37]
**Virga et al.**	**39**	24 months	DCB	70.3	29.7	21.6	Superficial femoral		0	100	[38]
**Milnerowicz et al.**	**74**	12 months	RA + DCB	79.5	20.5	5.5	iliac and/or infrainguinal arteries		1.4	89	[39]
**Giannopoulos et al.**	**27**	12 months	Turbo -Power + DCB	90.8	9.2	9.1	Femoropopliteal		MALE: major adverse limb events 5.1	no significant (*p* = 0.170) difference between the 2 groups	[40]
	**51**	12 months	LA + PTA	59.9	40.1	44.3			MALE: major adverse limb events 3.7		
**Thieme et al.**	**89**	24 months	DCB	66	34	14,5	Superficial femoral		Freedom from TVR, major amputation, device- and procedure-related death: 82	76	[41]
**Shammeri et al.**	**26**	12 months	VSG	81.4, 36 m	18,6	25	Femoropopliteal	25	7.69		[42]
**RELINE/Bosiers et al.**	**39**	12 months	VSG (CSD)	74.8	25.2	20.1	Superficial femoral			93,6	[43]
	**44**	12 months	PTA	28	72	57.8	Superficial femoral			87.8	
**ZILVER PTX/Zeller et al.**	**108**	12 months	DES	78.8	21.2	19, 12 m/60.8, 24 m	Femoropopliteal		0	60.9, 24 m	[44]
**Murato et al.**	**57**	12 months	DES	49	51	not available	Femoropopliteal	44.1	MALE: major adverse limb events: 25.5	not available	[45]
	**44**	12 months	PTA	14	86	not available		90.3	MALE: major adverse limb events 53.6	not available	
**Van Den Berg et al.**	**14**	12 months	Laser + DCB	91.7	8,3	7	Infrainguinal arteries		7	improved in all patients	[46]
**PERMIT-ISR Trial**	**59**	24 months	MATH + DCB	82.5	17.5	15.3	Superficial femoral and proximal popliteal artery	0	1	the ABI changed at 12 months were significantly improved from baseline (*p* < 0.01)	[47]
**Tomoi Y. et al.**	**28**	36 months	POBA + DES		32.2% 24 m	15.3	Femoropopliteal	18.4 24 m	MALE: 15.3 at 24 m	60.7	[48]
**Zhang B. et al.**	**28**	14 months	Debulking (if nedeed) + POBA + DCB	79.2, 14 m		8.5, 14 m	Femoropopliteal	0	0	Clinical symptoms improved by at least 1 Rutherford category in 82.1% of limbs	[49]
**Bosiers M. et al.**	**39**	24 months	VSG	74.8 at 12 m/58.40 at 24 m		33.7 at 24 m	Femoropopliteal	0	0	Clinical symptoms improved by at least 1 Rutherford 93.50% at 12 m and 93.10% at 24 m	[50]

Duration of follow-up is shown in the table, and the results are related to this follow-up, unless a different follo-up duration is indicated (in month). RCT = randomized control trial, PSMs = propensity score matched study, MPRT = multicenter prospective randomized controlled trial, NRSO = not randomized observational studies, PEB = paclitaxel eluting balloon, EES = everolimus eluting stent, PES = paclitaxel eluting stent, DES = drug-eluting stent, DCB = drug coated balloon, BMS = bare metal stent, LD = laser debulking, PTA = percutaneous transluminal angioplasty, POBA = plain old balloon angioplasty, LA = laser atherectomy, RA = rotational atherectomy, VSG = Viabahn Stent-Graft, CSD = covered stend deployment, RS = retrospective study, PMT = prospective multicenter trial, DCBA = drug coated balloon angioplasty, DA = directional atherectomy, MATH = percutaneous mechanical atherectomy plus thrombectomy.

## Data Availability

Not applicable.

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
