# Peer review of "The Diagnostic Value of Circulating Biomarkers and Role of Drug-Coated Balloons for In-Stent Restenosis in Patients with Peripheral Arterial Disease"

_diagnostics, 2022, doi:10.3390/diagnostics12092207_

Round 1
Reviewer 1 Report
Nicely written review.
Author Response
Thank you for your comment and your appreciation.
Reviewer 2 Report
Montelione et al. prepared an interesting review paper on a very important subject such as in-stent restenosis after endovascular treatment of peripheral arterial disease. The issue is significant because cardiovascular diseases are the leading cause of mortality and morbidity worldwide. Peripheral arterial disease is, next to coronary heart disease and cerebrovascular disease, a clinical manifestation of atherosclerotic cardiovascular disease. Although the paper is generally very well prepared and, in my opinion, should be considered for publication in Diagnostics, some modifications are needed to improve the quality of the presented manuscript further.
Major revisions:
1) The use of the abbreviation BA (balloon angioplasty) is not necessary because the Authors already have introduced the abbreviation PTA (percutaneous transluminal angioplasty). PTA is exactly balloon angioplasty. When we think about subsequent stent implantation, we use the expression “PTA with stenting” or “PTA with stent implantation”.
2) The “Introduction” should be significantly changed. First of all, it should be mentioned that chronic lower extremity ischemia is the most important but not the only manifestation of PAD. Moreover, when we talk about chronic lower extremity ischemia, not only femoropopliteal stenosis or occlusion are of interest, but also the aortoiliac and below-the-knee segments. Moreover, not always stent implantation is necessary in the endovascular treatment of PAD because, in some cases, PTA is sufficient. The term critical limb ischemia should be used and elucidated. It should be mentioned that diabetes significantly influences the course of PAD because, in patients with diabetes, the arteries below the knee are more often affected, and stenoses and occlusions are more often multilevel. Diabetes is generally associated with different pathological mechanisms influencing cardiovascular disease pathogenesis, such as specific lipid disorders. (see for example the following recently published references: 10.3390/antiox11050856, 10.3390/ijerph17249339, 10.1001/jama.2021.2126, 10.1016/j.tcm.2019.04.002, 10.1177/15443167221095175).
3) The database search methodology is well described. With such a precisely defined methodology, is it really a narrative review rather than a systematic review or an intermediate form between systematic and narrative review? I have my doubts. Please think about it.
4) Tables take up a lot of space and are quite difficult to read. Moreover, there are hardly any conclusions that could be drawn from these tables. The results of the review are in chapter 3, where the research is tabulated, while in chapter 4, the text is actually devoted to a different subject, using different literature, so it is not really a discussion of the research from chapter 3. In my opinion, the main text should include only some basic tabulation of the research collected in chapter 3 with a verbal commentary, while the full tables as presented now should be found in supplementary materials. Please consider it.
5) The chapter diagnostic implications (chapter 4) should be broadened and organized. I propose to separate a subsection on inflammatory parameters, a subsection on microRNA and possible further subsections for additional markers.
Minor revisions:
1) The references should be marked in the main text in square brackets, e.g. [1], [2], etc.
2) In the list of abbreviations below the tables, one of the characters should be consistently used, i.e.: or =.
Author Response
Reviewer 2
Montelione et al. prepared an interesting review paper on a very important subject such as in-stent restenosis after endovascular treatment of peripheral arterial disease. The issue is significant because cardiovascular diseases are the leading cause of mortality and morbidity worldwide. Peripheral arterial disease is, next to coronary heart disease and cerebrovascular disease, a clinical manifestation of atherosclerotic cardiovascular disease. Although the paper is generally very well prepared and, in my opinion, should be considered for publication in Diagnostics, some modifications are needed to improve the quality of the presented manuscript further.
Major revisions:
- The use of the abbreviation BA (balloon angioplasty) is not necessary because the Authors already have introduced the abbreviation PTA (percutaneous transluminal angioplasty). PTA is exactly balloon angioplasty. When we think about subsequent stent implantation, we use the expression “PTA with stenting” or “PTA with stent implantation”
Thank you for your comment. The BA abbreviation was deleted
- The “Introduction” should be significantly changed. First of all, it should be mentioned that chronic lower extremity ischemia is the most important but not the only manifestation of PAD. Moreover, when we talk about chronic lower extremity ischemia, not only femoropopliteal stenosis or occlusion are of interest, but also the aortoiliac and below-the-knee segments. Moreover, not always stent implantation is necessary in the endovascular treatment of PAD because, in some cases, PTA is sufficient. The term critical limb ischemia should be used and elucidated. It should be mentioned that diabetes significantly influences the course of PAD because, in patients with diabetes, the arteries below the knee are more often affected, and stenoses and occlusions are more often multilevel. Diabetes is generally associated with different pathological mechanisms influencing cardiovascular disease pathogenesis, such as specific lipid disorders. (see for example the following recently published references: 10.3390/antiox11050856, 10.3390/ijerph17249339, 10.1001/jama.2021.2126, 10.1016/j.tcm.2019.04.002, 10.1177/15443167221095175).
Thank you for your comment. The Introduction section was changed as suggested.
- The database search methodology is well described. With such a precisely defined methodology, is it really a narrative review rather than a systematic review or an intermediate form between systematic and narrative review? I have my doubts. Please think about it.
Thank you so much for your consideration. But as pointed out, the intent of the manuscript is to provide a narrative review. This sentence was added to the discussion:
“As anticipated in the introduction, this manuscript has the intention of providing a narrative review on the current knowledge on the value of circulating biomarkers as predictors of ISR and to foster the scientific debate on the advantages of using DCB in the treatment of ISR in the FP district. In fact, the use of the stent in the FP district will become an increasingly pursued practice in the field of vascular surgery, making it increasingly necessary to study in deep this knowledge”.
4) Tables take up a lot of space and are quite difficult to read. Moreover, there are hardly any conclusions that could be drawn from these tables. The results of the review are in chapter 3, where the research is tabulated, while in chapter 4, the text is actually devoted to a different subject, using different literature, so it is not really a discussion of the research from chapter 3. In my opinion, the main text should include only some basic tabulation of the research collected in chapter 3 with a verbal commentary, while the full tables as presented now should be found in supplementary materials. Please consider it.
Thank you for your comment. A verbal commentary of the results reported in Table 3 was added to the text with a section in the results and another in the discussion. If you find it useful, tables 1 and 2 could be inserted as supplementary materials.
5) The chapter diagnostic implications (chapter 4) should be broadened and organized. I propose to separate a subsection on inflammatory parameters, a subsection on microRNA and possible further subsections for additional markers.
Thank you for your comment. The section was broadened and organized as suggested.
Minor revisions:
- The references should be marked in the main text in square brackets, e.g. [1], [2], etc.
Thank you for your comment. The square brackets were added to the text.
2) In the list of abbreviations below the tables, one of the characters should be consistently used, i.e.: or =.
Thank you for your comment. The list abbreviation was corrected.
Reviewer 3 Report
In a narrative review, the authors aimed to summarize the current knowledge on the drug-coated balloon (DCB) on in-stent restenosis (ISR) and to identify the advantages of using DCB in the treatment of ISR in the femoropopliteal district. It is suggested that the authors consider a few comments.
- The title includes “Diagnostic value of circulating biomarker”; however, it has not been mentioned as the study question in the introduction. In addition, the literature review on the diagnostic value of biomarkers is sparse and limited to a few numbers of studies.
- Lack of novelty is a major concern with the present manuscript. The role of DCBs in in-stent restenosis and PAD patients has been discussed in several review articles with more detailed and more conclusive results. The authors therefore should explain the novel aspects of their study and justify the need for it in current literature.
- Databases used for the present systematic review are limited. The authors might have benefited from including major databases including Web of Science, ProQuest, ScienceDirect, etc.
- In several included studies, patients have not received DCBs as part of their treatment but they have been assessed in the present narrative study on DCB. This issue needs clarification.
- In discussion, several studies need appropriate citation.
- The manuscript needs English Editing.
Author Response
Reviewer 3
In a narrative review, the authors aimed to summarize the current knowledge on the drug-coated balloon (DCB) on in-stent restenosis (ISR) and to identify the advantages of using DCB in the treatment of ISR in the femoropopliteal district. It is suggested that the authors consider a few comments.
Thank you for your comment. This comment was added to the text “As anticipated in the introduction, this manuscript has the intention of providing a narrative review on the current knowledge on the value of circulating biomarkers as predictors of ISR and to foster the scientific debate on the advantages of using DCB in the treatment of ISR in the FP district. In fact, the use of the stent in the FP district will become an increasingly pursued practice in the field of vascular surgery, making it increasingly necessary to study in deep this knowledge”.
- The title includes “Diagnostic value of circulating biomarker”; however, it has not been mentioned as the study question in the introduction. In addition, the literature review on the diagnostic value of biomarkers is sparse and limited to a few numbers of studies.
Thank you for your comment. The introduction was revised and the question of “circulating biomarkers” was more underlined.
- Lack of novelty is a major concern with the present manuscript. The role of DCBs in in-stent restenosis and PAD patients has been discussed in several review articles with more detailed and more conclusive results. The authors therefore should explain the novel aspects of their study and justify the need for it in current literature.
Thank you for your comment. However, we believe that the topic discussed in this review still hides multiple factors that are not considered today. It is therefore essential to undertake a comprehensive process that takes into account the advantages and disadvantages of using DCBs in the treatment of the ISR. Furthermore, if we consider the advantages of a possible detection of circulating biomarkers that could predict the risk of the ISR the importance of studies like this is there for all to see.
- Databases used for the present systematic review are limited. The authors might have benefited from including major databases including Web of Science, ProQuest, ScienceDirect, etc.
Thank you for your comment. However, confident of the research method used, we believe that Pubmed and Scopus can represent with absolute respect most of the studies on this topic, not risking to leave out relevant reports.
- In several included studies, patients have not received DCBs as part of their treatment but they have been assessed in the present narrative study on DCB. This issue needs clarification.
Thank you for your comment. However, we decided to consider studies that evaluated the treatment of ISR in the FP district, including these that did not involve the use of DCB, but the use of alternative methods such as conventional PTA or atherectomy. We believe that this approach could help to better compare the results of the use of DCB, emphasizing its advantages in the treatment of the ISR.
- In discussion, several studies need appropriate citation.
Thank you for your comment. The discussion was expanded and the reference list was updated with appropriate citation.
- The manuscript needs English Editing.
Thank you for your comment.
Round 2
Reviewer 2 Report
Thank you for inviting me to review the revised version of the work. It is good that the authors take up an important topic in their scientific work, such as improving the effectiveness of revascularization in patients with peripheral arterial disease. After reading the revised version of the work, I regret to say that the authors did not satisfactorily address the comments contained in my review and that of other reviewers, and the changes introduced are too small. I lack a certain consistency and signs of originality in this work. Of course, the review by assumption does not bring new results, but it assumes that the already available information will be analyzed and presented in an original way, throwing out some new thought on a given topic. My objection is that most of the work consists of tables, the content of which does not add anything new to the publications that are cited in them. I do not see any interesting conclusions being drawn in the discussion, or that new research directions have been proposed. I appreciate the authors' efforts, but the work in its current version cannot be published in a journal with Impact Factor. After comparing the two versions of the manuscript that I reviewed, I see no further possibility for this manuscript to be satisfactorily improved. Please prepare the work again and submit a new manuscript.
Author Response
Thank you for your considerations.
We have further expanded the biomarker section by adding a figure on the diagnostic implications of inflammatory markers and microRNAs in intrastent restenosis and expanded the text in the "Introduction" and "Discussion" sections.
I hope that the manuscript can now meet the reviewers' requests and be considered for publication.
Round 3
Reviewer 2 Report
Thank you for inviting me to a review of the revised version of the work. Unfortunately, I still believe that the current version of the manuscript is ineligible for publication in a journal with such a good Impact Factor as the journal Diagnostics. A significant part of the work is still made up of extensive tables, which show little and are not readily readable. The text does not contain any interesting conclusions, it does not satisfactorily indicate the directions of further research. The topic undertaken by the authors is important, interesting and should be developed, but unfortunately this work is not suitable for publication in a good journal.